METHODS

# MIRAGE: Robust multi-modal architectures translate fMRI-to-image models from vision to mental imagery

Reese Kneeland[1,2*☯] Cesar Kadir Torrico Villanueva[2☯], Tong Chen[2,3☯], Jordyn Ojeda[1], Shubh Khanna[4], Jonathan Xu[2,5], Paul S. Scotti[2,6,7], Thomas Naselaris[1]

1 University of Minnesota, Minneapolis, Minnesota, United States of America, 2 Medical AI Research Center (MedARC), San Francisco, California, United States of America, 3 University of Sydney, Sydney, Australia, 4 Stanford University, Palo Alto, California, United States of America, 5 Alljoined, San Francisco, California, Unites States of America, 6 Sophont, San Francisco, California, Unites States of America, 7 Princeton Neuroscience Institute, Princeton, New Jersey, United States of America

☯ Core contribution.
* reesekneeland@gmail.com

## Abstract

To be useful for downstream applications, vision decoding models that are trained to reconstruct seen images from human brain activity must be able to generalize to internally generated visual representations, i.e., mental images. In an analysis of the recently released NSD-Imagery dataset, we demonstrated that while some modern vision decoders can perform quite well on mental image reconstruction, some fail, and that state-of-the-art (SOTA) performance on seen image reconstruction is no guarantee of SOTA performance on mental image reconstruction. Motivated by these findings, we developed **MIRAGE**, a method explicitly designed to train on vision datasets and cross-decode mental images from brain activity. **MIRAGE** employs a linear backbone and multi-modal text and image features as input to a diffusion model. Feature metrics and human raters establish MIRAGE as SOTA for mental image reconstruction on the NSD-Imagery benchmark. With ablation analysis we show that mental image reconstruction works best when decoders use image features with relatively few dimensions and include guidance from text-based and both high- and low-level image-based features. Our work indicates that–given the right architecture–existing large-scale datasets using external stimuli are viable training data for decoding mental images, and warrant optimism about the future success and utility of mental image reconstruction.

### Author summary

Recent research has focused on developing "vision decoding models" that reconstruct images a person is currently viewing. While scientifically impressive, we argue that the tremendous potential of this technology lies in externalizing

**Data availability statement:** NSD-Imagery data has been released publically, and can be accessed at https://naturalscenesdataset. org/ The code for the MIRAGE method is open source and available at https://github.com/ MedARC-AI/MIRAGE.

**Funding:** This work was supported by the National Institutes of Health (R01EY023384 and R01EY038606 to TN) The funders had no role in study design, data collection and analysis, decision to publish, or preparation of the manuscript.

**Competing interests:** The authors have declared that no competing interests exist.

mental images—the private, internal world of our thoughts—rather than simply decoding what is already visible to the naked eye. This capability is a necessary step towards future applications, such as diagnostic instruments and communication tools for patients with disorders of communication or consciousness. In this study, we demonstrate that state-of-the-art vision decoding models are often not state-of-the-art when it comes to decoding mental imagery, and that the field's current focus on complex architectures optimized for vision has led to models that are often too fragile to capture the fainter, noisier signals of the imagination. We introduce MIRAGE, a method explicitly designed to bridge this gap by prioritizing robustness over complexity. Our findings indicate that making progress toward practical brain-computer interfaces requires a shift in focus: researchers must target mental imagery directly rather than assuming vision-based performance will translate in the service of impactful downstream use-cases.

## 1. Introduction

The ability to decode and reconstruct mental images—internally generated visual representations not driven by sensory input—from brain activity has tremendous potential for downstream applications such as brain-computer interfaces and medical diagnostics for patients with disorders of communication or consciousness. Externalizing mental images also provides insights into cognitive processes that would be otherwise inaccessible.

Recent research on decoding has focused on developing "vision decoding models" that are trained and tested on brain responses to seen images. To utilize such vision decoding models for the downstream applications we envision, it is important to demonstrate that these vision decoding models generalize well when tested on mental imagery. An evaluation of a suite of high-performing vision decoding models on the recently-released NSD-Imagery dataset [1]–which contains samples of brain activity patterns measured with 7T fMRI in human subjects as they generate mental images–has revealed that decoding performance on seen images is a poor predictor of decoding performance on mental images: while some of the vision decoders evaluated performed quite well on mental image reconstruction, some failed, and state-of-the-art (SOTA) performance on seen image reconstruction was no guarantee of SOTA performance on mental image reconstruction. This indicates that making progress toward a decoder with practical utility will require careful consideration of the factors that make mental image reconstruction challenging, and of the attributes that make vision decoders more or less likely to generalize. To that end, we present the following contributions:

1. We introduce **MIRAGE** (**M**ental **I**mage **R**econstruction using **A**dvanced **G**enerative Mod**E**ls), demonstrating that the principled integration of linear decoding backbones with low-dimensional multi-modal feature spaces is an effective technique for enabling cross-decoding of internally generated mental images from vision-only training data.

2. We establish **MIRAGE** as a SOTA mental imagery decoding method by comparing evaluations across a broad selection of image feature metrics and human preference ratings derived from large-scale behavioral experiments.

3. We conduct a detailed ablation analysis to provide empirical evidence for the architectural choices that facilitate generalization from seen to mental image reconstruction, and identify the specific technical reasons why MIRAGE successfully generalizes to NSD-Imagery where MindEye2 and other more complex vision decoding models fail (Fig 1).

## 1.1. Related work

[3] Basic neuroscience has demonstrated an extensive overlap in the representation of seen and mental images [4–8]. Nonetheless, differences between vision and mental imagery make cross-decoding from vision to imagery challenging [9,10]. Compared to vision, brain activity during mental imagery has a lower signal-to-noise ratio (SNR) [11], varies along fewer signal dimensions [12], and encodes imagined stimuli with less spatial resolution [13,14].

Previous studies have reported successful *n*-way classification of mental images [9,15,16], retrieval of mental images of natural scenes using a visual encoding model [7,17], and reconstruction of simple blobs, letters, or singular natural objects [6,18–22]. With the open releases of CLIP [23], Stable Diffusion [24], and large-scale fMRI datasets like NSD [25], newer vision decoding models now yield highly accurate reconstructions of natural scenes [26–36]. These methods map fMRI brain activity patterns to embeddings of pre-trained deep learning models that are used to drive a diffusion model [2,37–39] to generate image reconstructions of the content present in visual cortex.

Tests of open-source vision decoding models [2,21,26–29] on mental imagery activity in the NSD-Imagery dataset [1] showed that improved performance on vision decoding does not necessarily translate to mental imagery decoding. For example, MindEye2 [2] was the SOTA model on the test set of NSD, but not on the imagery trials of NSD-Imagery (Fig 1). Furthermore, in that study it was observed that methods with linear backbones, compact representations, and multimodal guidance yielded the best performance. We interpreted these observations as follows: First, [40] showed that linear mappings from CLIP to brain activity explains much of variance in activity in visual cortex. Thus, we should expect linear methods to work quite well when decoding CLIP. Second, as models increase in complexity, the risk of overfitting to noise in the input increases; this danger is especially acute in our case, given that imagery has very low SNR. Thus, representations with fewer dimensions are to be favored, all else being equal. Third, imagery and visual activity are most aligned in brain areas with language-like representations ([3,9,13]). Thus, it would make sense to include high-quality language-like representations in the decoding pipeline for mental imagery. These ideas informed the design of MIRAGE.

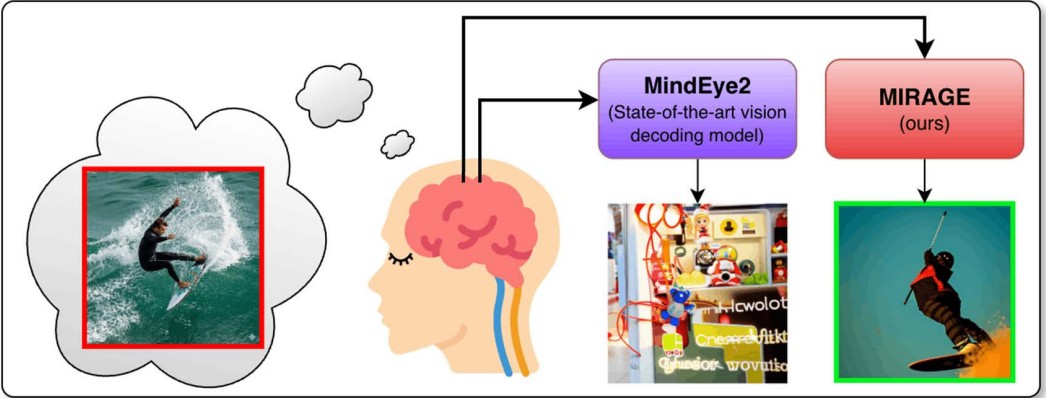

**Fig 1. MIRAGE (ours) vs MindEye2 [2] reconstructions of an imagined image from fMRI brain activity.**

Our results, including detailed ablation analyses, support our reasoning, and explain why MIRAGE is SOTA for mental image reconstruction.

## 2. Results

To generate baseline reconstructions for the NSD-Imagery benchmark, we utilized the official open-source implementations provided by the authors of each method, and sampled 10 reconstructions from the posterior distribution of each model for our analysis. The mental image reconstructions produced by **MIRAGE** (Fig 2 and H) are noticeably more faithful to the ground truth images the subjects were instructed to imagine than Brain Diffuser (the previous SOTA model applied to NSD-Imagery), and the other methods we evaluate against. For simple stimuli, the overall structure and orientation of the ground truth images are noticeably improved. For complex stimuli, we successfully reconstruct images that contain similar categories and themes—such as donuts, a man riding a surfboard, an animal with a beak—and with notably improved structural accuracy for complex objects such as the pose of the surfer and the position of the donuts. For conceptual stimuli, the images reconstructed clearly reflect content that is related or identical to the corresponding concept word, with a zebra for "zebra", a cat for "mammal", and a recognizable banana for the "banana" prompt.

Vision reconstructions for simple and complex stimuli (Fig B and G in S1 Text) also perform well, despite vision decoding performance not being our primary target. The simple stimuli reconstructions are less visually distorted as a result of our low-level guidance, and the complex stimuli hold up well to previous methods. Median and worst-case reconstructions are in Fig C, D, E, F, I, J, K, and L in S1 Text.

### 2.1. Feature metric evaluations

We provide performance benchmarks against existing methods evaluated on NSD-Imagery [1] using the metrics in Table 1. For all methods, we output 10 reconstructions per test sample from each method and report averaged metrics across them. Metrics can fluctuate due to the stochastic nature of fMRI-to-image models, and this averaging step increases the reliability of results. Statistical significance measures can be found in Table A in S1 Text. Across the majority of metrics, **MIRAGE** shows SOTA performance on mental image reconstructions. We note that although these metrics are often used as a proxy for human judgment, many research efforts have established that these metrics do not closely approximate or align with human assessments of content [49] or quality [50]. We observe that they are particularly volatile with a dataset as small as NSD-Imagery. For this reason, we provide extensive behavioral evaluations of our results by human raters in Section 2.2. Benchmarks separated by stimulus type are provided in Tables C and D in S1 Text. Results of our method on the NSD shared1000 test set are also provided in Table B in S1 Text.

### 2.2. Human ratings of reconstruction quality

To ensure downstream applicability, reconstruction methods must produce outputs that are meaningful to human observers. Although we report standard image feature metrics, prior work indicates that these automated scores often dissociate from human perceptual judgments of quality and semantic content [2,49,50]. These factors, in addition to the high volatility of automated metrics on small datasets such as NSD-Imagery, lead us to treat human evaluation as the definitive standard for assessing model performance. We thus conducted several large-scale online behavioral experiments in which human raters (n = 500) assessed the quality of the reconstructions (See Appendix A.14 in S1 Text).

**Experiment 1** To perform a systematic controlled experiment analyzing our results, we had human raters perform a 2-alternative forced choice (2AFC) judgment about whether a reconstruction was more similar to the ground truth image than a randomly selected "distractor" reconstruction of a brain activity pattern originating from a different stimulus that was sampled from the same stimulus type, method, and NSD subject. Results (Table 2) confirm **MIRAGE** as SOTA for every stimulus type ($p < 0.001$) (Fig 3).

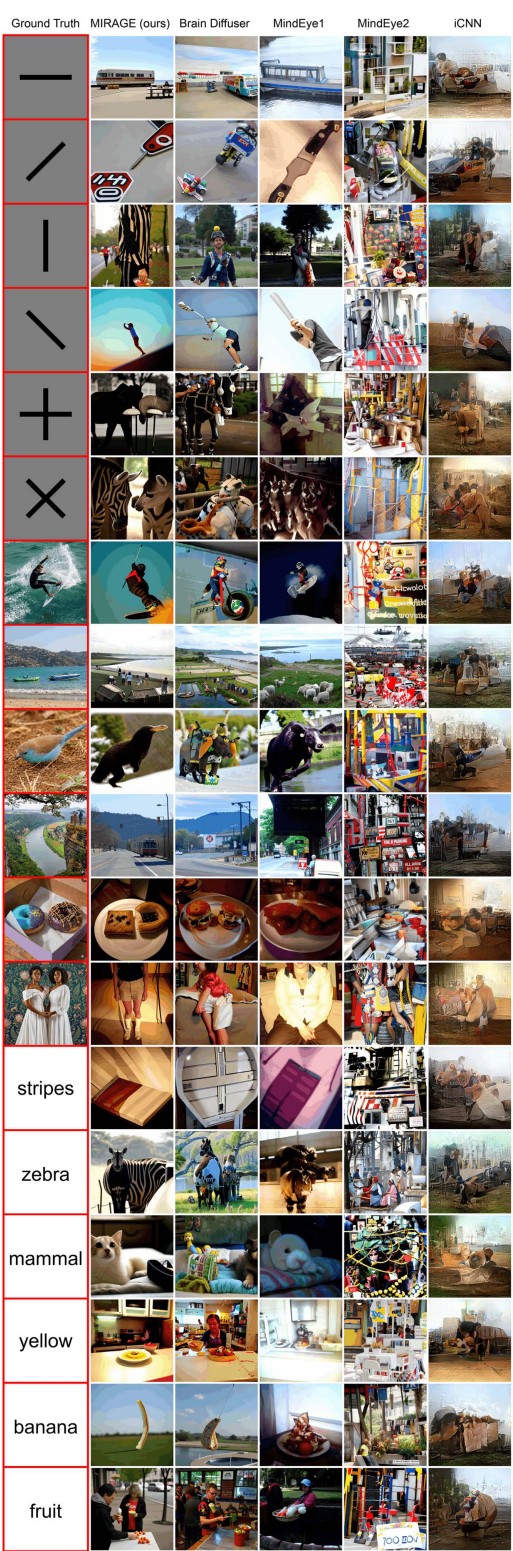

**Fig 2. Qualitative comparison of reconstruction methods on imagined stimuli from NSD-Imagery.** Reconstructions selected are the outputs sampled from each method and stimulus with the highest scores on all of the image feature metrics in Table 1. Examples from more methods can be seen in Appendix A.5 in S1 Text.

**Table 1.** Quantitative comparison between reconstruction methods for both imagery and vision trials on simple and complex stimuli (conceptual stimuli have no ground truth images). PixCorr is the pixel-level correlation score. SSIM is the structural similarity index metric [41]. AlexNet(2) and AlexNet(5) are the 2-way comparisons (2WC) of layers 2 and 5 of AlexNet [42]. CLIP is the 2WC of the output layer of the CLIP ViT-L/14 Vision model [23]. Inception is the 2WC of the last pooling layer of InceptionV3 [43]. EffNet-B and SwAV are distance metrics gathered from EfficientNet-B13 [44] and SwAV-ResNet50 [45] models. Each brain correlation score is calculated using voxels from within the respective regions of the visual cortex. For EffNet-B and SwAV distances, lower is better. For all other metrics, higher is better. Bold indicates best performance, and underlines second-best performance. Additional details on the metrics used, including explanations of 2-way comparisons and brain correlation scores, are in Appendix A.6 in S1 Text. A breakdown of model performance across the different types of stimuli is in Appendix A.9 in S1 Text. Details for our implementation of iCNN are provided in Appendix A.15 in S1 Text.

| Method | Low-Level | | | | High-Level | | | | Brain Correlation | | | Captions | | | |
| --- | --- | --- | --- | --- | --- | --- | --- | --- | --- | --- | --- | --- | --- | --- | --- |
| | Pix-Corr ↑ | SSIM ↑ | Alex(2) ↑ | Alex(5) ↑ | Incep ↑ | CLIP ↑ | Eff ↓ | SwAV ↓ | Early Vis. ↑ | Higher Vis. ↑ | Visual Cortex ↑ | ROUGE-L ↑ | METEOR ↑ | Sentence ↑ | CLIP-L ↑ |
| **Mental Imagery Reconstructions** | | | | | | | | | | | | | | | |
| **MIRAGE (ours)** | 0.104 | 0.398 | **63.92%** | **62.46%** | 52.25% | **57.46%** | **0.914** | 0.575 | **0.204** | 0.142 | **0.168** | **0.154** | **0.095** | **0.176** | 0.469 |
| MindEye1 [29] | 0.086 | 0.349 | 59.56% | 61.00% | 52.03% | 54.72% | 0.948 | 0.564 | 0.180 | 0.135 | 0.155 | – | – | – | – |
| Brain Diffuser [28] | 0.064 | 0.401 | 52.14% | 58.35% | **52.73%** | 54.07% | 0.935 | 0.585 | 0.133 | 0.127 | 0.141 | – | – | – | – |
| iCNN [21] | **0.108** | 0.340 | 50.57% | 55.25% | 49.39% | 41.72% | 0.994 | **0.560** | 0.113 | 0.062 | 0.081 | – | – | – | – |
| MindEye2 [2] | 0.036 | **0.414** | 47.60% | 55.38% | 46.02% | 50.78% | 0.966 | 0.591 | 0.069 | 0.055 | 0.061 | 0.143 | 0.080 | 0.162 | **0.484** |
| MindBridge [46] | 0.030 | 0.200 | 44.13% | 52.46% | 45.55% | 49.70% | 0.980 | 0.627 | 0.079 | 0.079 | 0.075 | – | – | – | – |
| NeuroPictor [47] | 0.022 | 0.305 | 43.18% | 44.85% | 44.15% | 46.40% | 0.994 | 0.612 | 0.084 | 0.054 | 0.140 | – | – | – | – |
| BrainRAM [48] | 0.056 | 0.372 | 51.78% | 56.29% | 52.52% | 53.73% | 0.927 | 0.577 | 0.139 | **0.145** | 0.112 | – | – | – | – |
| **Vision Reconstructions** | | | | | | | | | | | | | | | |
| **MIRAGE (ours)** | 0.221 | 0.442 | **79.03%** | 76.57% | **69.75%** | **66.69%** | **0.879** | 0.546 | 0.363 | **0.262** | 0.316 | 0.157 | 0.105 | 0.216 | 0.469 |
| MindEye1 [29] | 0.218 | 0.412 | 73.56% | 80.81% | 62.44% | 65.34% | 0.881 | **0.510** | 0.374 | 0.253 | 0.311 | – | – | – | – |
| Brain Diffuser [28] | 0.107 | 0.455 | 60.34% | 72.84% | 60.95% | 58.31% | 0.908 | 0.555 | 0.247 | 0.229 | 0.255 | – | – | – | – |
| iCNN [21] | **0.224** | 0.385 | 71.67% | **81.35%** | 61.16% | 49.03% | 0.926 | 0.524 | **0.442** | 0.246 | 0.338 | – | – | – | – |
| MindEye2 [2] | 0.161 | **0.480** | 70.10% | 77.52% | 62.69% | 65.93% | 0.886 | 0.512 | 0.352 | 0.237 | 0.290 | **0.171** | **0.118** | **0.249** | **0.515** |
| MindBridge [46] | 0.117 | 0.352 | 58.47% | 70.76% | 58.83% | 64.49% | 0.915 | 0.565 | 0.245 | 0.227 | 0.232 | – | – | – | – |
| NeuroPictor [47] | 0.055 | 0.364 | 62.27% | 66.42% | 49.92% | 53.49% | 0.949 | 0.571 | 0.272 | 0.192 | **0.363** | – | – | – | – |
| BrainRAM [48] | 0.097 | 0.409 | 63.11% | 67.99% | 58.50% | 59.79% | 0.894 | 0.530 | 0.215 | 0.226 | 0.175 | – | – | – | – |

**Experiment 2** Human raters viewed a ground truth image, a reconstruction of that stimulus from a vision trial, and a reconstruction of the same stimulus from an imagery trial. Raters provided continuous measures of similarity between each reconstruction and the ground truth image. The rating provided a direct comparison of the similarity between vision and imagery reconstructions. **MIRAGE** shows SOTA generalization from vison to imagery [40].

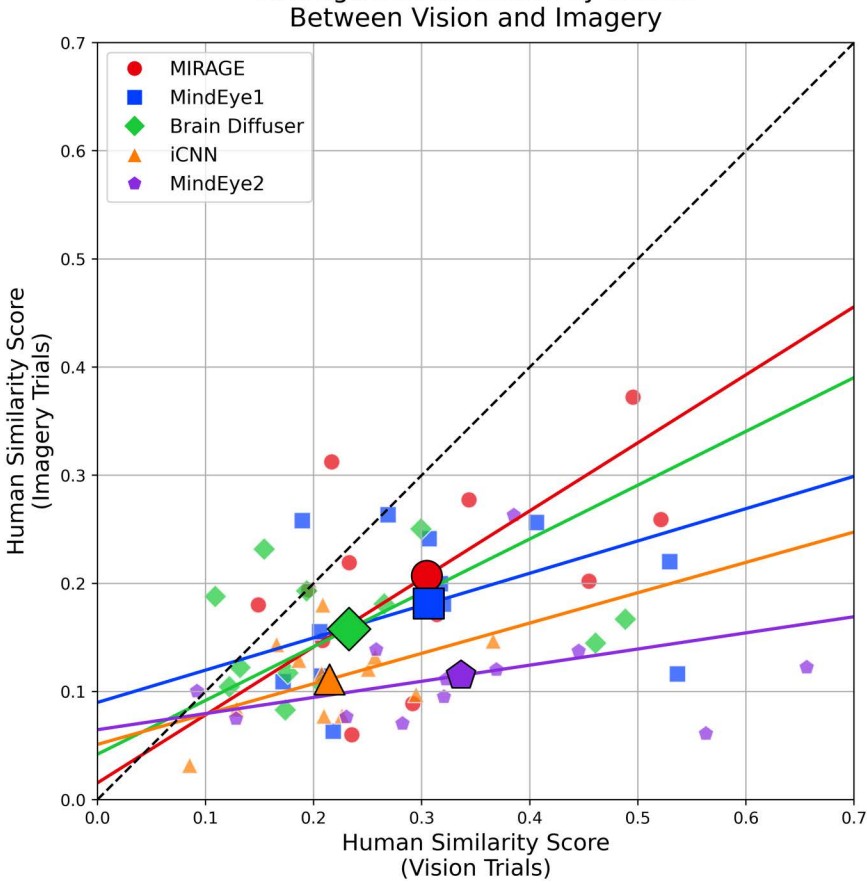

**Fig 3.** Human similarity scores for simple and complex stimuli: X-axis = vision, Y-axis = imagery; each point is the mean over 12 samples (larger bold points are the overall means), colored/shaped by method. PCA-fit slopes closer to unity indicate tighter imagery–vision correspondence; dashed unity line shown.

**Experiment 3** Mental image reconstructions of the conceptual stimuli are particularly difficult to evaluate, as they do not have associated ground truth images or a meaningful match to vision trials. To compare the performance of our method on these stimuli relative to other models trained on NSD, we conducted a third behavioral experiment that pitted reconstruction methods head-to-head by presenting human raters with a ground truth conceptual stimulus and two reconstructions of that stimulus sampled at random from the collection of methods we evaluate against in this work. Every trial in this experiment is a head-to-head comparison between two random reconstruction methods, and so over the course of the whole experiment, we gather lots of trials between all combinations of two methods. The "similarity score advantage" is the average difference between one method and another across the trials where both methods were presented. Fig 4A plots the average "similarity score advantage" for all combinations of methods. Reconstructions from our method are the most strongly preferred in head-to-head comparisons ($p < 0.001$) and have the largest advantage in all comparison cases evaluated (Table 2).

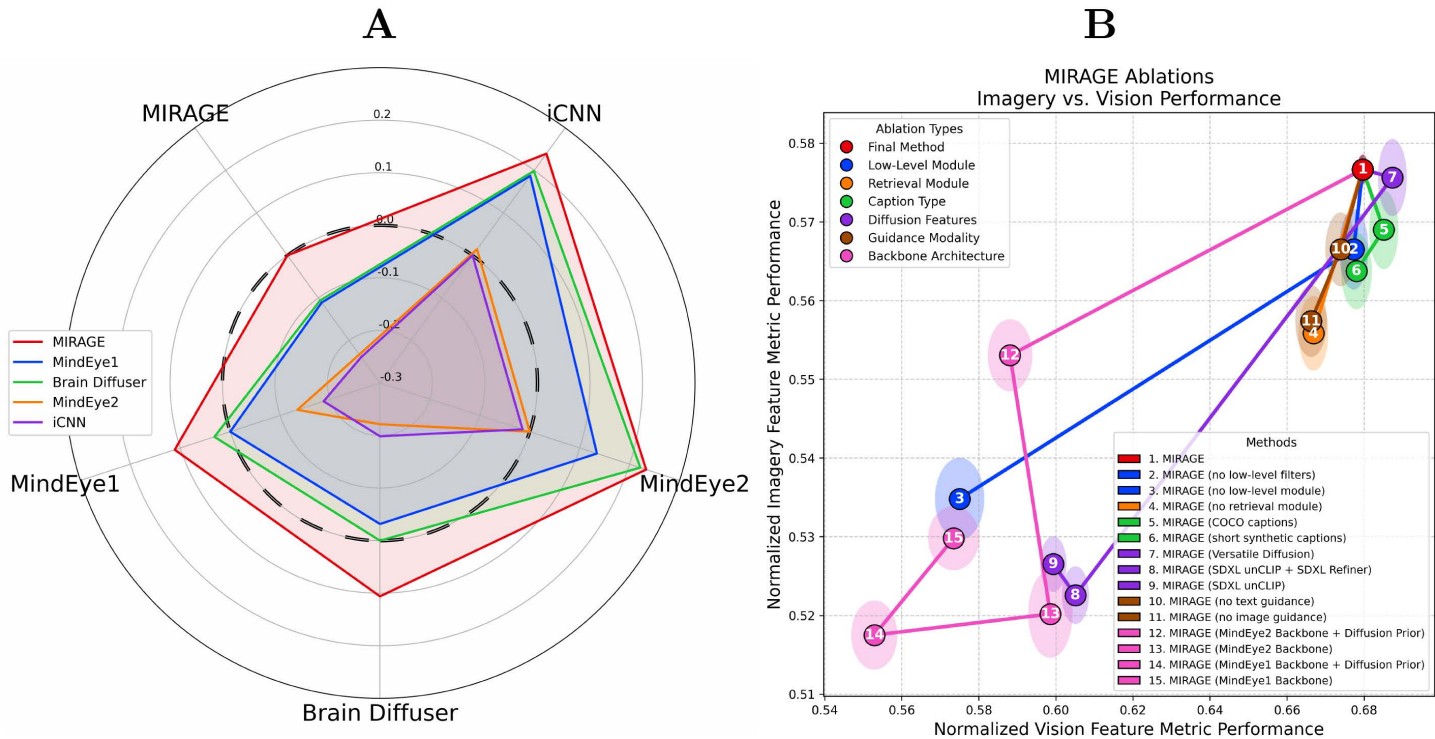

**Fig 4. (A)** Head-to-head human similarity score results for the conceptual stimuli.The Y-axis represents the similarity score advantage (difference between target method's score and the alternative, on the radial X-axis); a larger colored polygon area indicates a stronger advantage, and the dashed circle at unity denotes equal performance. MIRAGE outperforms all other methods ($p < 0.001$). **(B)** Ablation analyses: model variants (numbered circles) under each ablation type (color) are assessed via the normalized average of all feature metrics (Table 1), with vision on the x-axis and imagery on the y-axis. The standard error of each point along each axis is visualized with a shaded ellipse of the same color. For details on how we compute the normalized average metric scores, see Appendix A.6.1 in S1 Text.

## 2.3. Ablation study

We systematically ablated model components to identify which were most important for mental imagery reconstruction. Colored numeric identifiers refer to Fig 4B. **MIRAGE** is identified as (1). We also provide a hyperparameter search over the ridge parameter $\lambda$ in Appendix A.1 in S1 Text.

**Low-Level Module** We found that despite the relatively low spatial resolution of mental imagery, the low-level decoding module provides much of the needed structure to accurately reconstruct the target stimulus (3). We also observe that our image filtering technique induces a small performance increase for mental imagery reconstructions (2), suggesting that this technique partially mitigates the loss of structural detail in mental vs. seen image reconstructions.

**Retrieval Module** We see a notable boost from our retrieval procedure (4), demonstrating that high-dimensional image embeddings can be decoded and used to guide a retrieval module to improve performance, even if those same embeddings are not suitable to drive an image generator.

**Caption Types** We examine the results of using COCO captions (5), as well as a set of synthetic "short" captions (6) that aim to replicate the length of the COCO captions (the median word count is 8 vs 10 for COCO), and evaluate the quality of captions provided by LLaVA v1.5-13B. These short synthetic captions did not provide a boost over COCO, but the longer synthetic captions (average word count = 34) utilized in our final method (1) provided an improvement.

**Diffusion Features** We replaced the 1×768 image embeddings and 77×1280 text embeddings used to drive Stable Cascade with the features used to drive Versatile Diffusion in MindEye1 and Brain Diffuser (7) [39] (257×768 image

**Table 2.** Human identification accuracy scores for the vision and mental imagery trials of the NSD-Imagery benchmark. Scores are provided for each method and stimulus type; best values are bolded and second best underlined (chance=50%, all $p<0.001$).

**A**

**Human Identification Accuracy**

| Method | All Stimuli ↑ | Simple ↑ | Complex ↑ | Conceptual ↑ |
|---|---|---|---|---|
| **Mental Imagery Reconstructions** | | | | |
| **MIRAGE (ours)** | **78.30%** | **73.93%** | **83.19%** | **77.68%** |
| MindEye1 | 73.00% | 71.01% | 82.28% | 65.68% |
| Brain Diffuser | 73.95% | 68.20% | 82.70% | 71.01% |
| iCNN | 66.15% | 66.81% | 70.04% | 61.60% |
| MindEye2 | 56.96% | 50.21% | 64.83% | 55.74% |
| **Vision Reconstructions** | | | | |
| **MIRAGE (ours)** | 84.22% | **80.77%** | 87.66% | N/A |
| MindEye1 | **84.29%** | 79.32% | 89.32% | N/A |
| Brain Diffuser | 80.13% | 71.79% | 88.46% | N/A |
| iCNN | 74.52% | 75.85% | 73.19% | N/A |
| MindEye2 | 83.05% | 75.54% | **90.56%** | N/A |

embedding, 77×768 text embedding), and SDXL unCLIP [2], the embedding used to drive MindEye2 (8,9) (257×1664 image embedding). Our smaller image embedding and larger text embedding provided the most robust guidance for reconstructing mental images (1).

**Guidance Modality** For both vision and mental imagery, image-only guidance (10) to Stable Cascade afforded better performance than text-only guidance (11). In both cases, but especially for mental imagery, the combination of text and image guidance (1) provided a large performance boost, suggesting that multimodal guidance plays an important role for mental imagery reconstruction.

**Backbone Architecture** The MLP backbone and diffusion prior architectures of MindEye1 (14,15) and MindEye2 (12,13) perform worse than our ridge regression backbone (1) for both vision and mental imagery, suggesting that these architectures tend to overfit to the training data in the core NSD experiment. Evidently, the potential for increased expressivity afforded by an MLP is a disadvantage when attempting to generalize vision reconstruction performance to new sessions and stimulus types, or to mental imagery.

**Scaling Behavior** We assess the scaling behavior of **MIRAGE** and other decoding pipelines with respect to the number of trial repetitions folded into the brain activity averages (Appendix A.10 in S1 Text), and the number of training data hours (Appendix A.11 in S1 Text). In both cases, the scaling behavior of **MIRAGE** outperforms all other decoding pipelines.

**Contribution of the Natural Image Prior** To assess the relative contributions of the natural image prior on reconstruction performance, we implemented two systematic controls. First, in Experiment 1, distractor images were generated by decoding randomly selected brain activity patterns. Thus, both target and distractor outputs were equally shaped by the diffusion prior. The high identification accuracy observed ($p<0.001$; see Table 2) therefore reflects genuine signal decoding rather than the prior's semantic bias. Second, we assessed reconstruction performance across varying strengths of diffusion guidance. We found that reconstructions with minimal guidance (with nearly pure VDVAE decoding) remained highly identifiable, while increasing guidance strength primarily enhanced image quality rather than semantic content (Fig P in S1 Text), mitigating recent concerns about the authenticity of fMRI-to-image reconstructions [51,52].

## 3. Discussion

In this work, we addressed the challenge of generalizing fMRI decoding models from visual perception to mental imagery, investigating the specific architectural constraints required to robustly reconstruct internal mental states using models trained exclusively on seen images. Specifically, we sought to identify why state-of-the-art vision decoders fail on mental imagery, and how to bridge the signal-to-noise gap between these two modalities. **MIRAGE** improves mental imagery reconstruction over vision reconstruction pipelines shown to be SOTA on NSD shared1000 test samples (i.e., MindEye2 [2]) and on the NSD-imagery dataset (i.e., BrainDiffuser [28]). Ablation studies revealed that the keys to the success of MIRAGE are (1) a simple linear decoding backbone, (2) a reduction in the dimension of latent image representations, and (3) the inclusion of high-quality multi-modal guidance to the diffusion model.

### 3.1. Neuroscientific interpretations

Our ablation studies (Section 2.3) identify two critical factors for generalization: reduced feature dimensionality and the inclusion of text-based guidance. These empirical results align with known properties of the visual cortex. First, the success of lower-dimensional image embeddings likely reflects the reduced signal-to-noise ratio (SNR) and coarser spatial resolution of mental imagery compared to vision [12,13]. High-dimensional embeddings, while expressive for high-SNR vision trials, are empirically prone to overfitting and harming generalization to the noisier imagery signal.

Second, the efficacy of text guidance supports our hypothesis that mental imagery relies heavily on semantic representations. Previous work has demonstrated that natural language supervision improves encoding models of higher-level visual cortex [3], suggesting that these regions encode visual information in a format that is semantically aligned with language. Higher-level visual cortex is also the area of the brain that most heavily overlaps with activations created by mental imagery [13], and so by incorporating text embeddings, we speculate that MIRAGE taps into this semantic overlap, allowing the model to stabilize reconstructions even when fine-grained visual details in the brain signal are degraded.

### 3.2. Societal Impact

Our work is a necessary step towards applications of mental image reconstruction, including diagnostic instruments for psychiatric conditions [53] and disorders of consciousness [54–56], as well as expressive alternative communication methods for patients with traumatic brain injuries [57], amyotrophic lateral sclerosis (ALS) [58], and locked-in syndrome [59]. While a significant fraction of this communicative value could be provided by a "concept decoder" whose output is linguistic or otherwise compressed, however, visual representations of internal states could complement linguistic representations, for example by revealing how a specific concept or text prompt is visually interpreted. As the saying goes, 'a picture is worth a thousand words'. While our work makes no claims about being able to decode what is unique and idiosyncratic about a specific individual's mental imagery, it does represent a first step towards customized bespoke representations of internal states, and is a very literal attempt at this goal motivated by the overlap between representations of vision and mental imagery, and measured against a known ground truth visual stimulus image.

Of course, the development of this technology raises concerns about the potential for misuse [60]. We propose that when deployed in a clinical setting, brain decoding should be defined as a medical procedure that yields private health information and should therefore be subject to all relevant laws pertaining to patient consent, risk / benefit assessment, and the protection of privacy. In all other settings, it seems obvious that laws governing brain decoding should require informed consent and, where necessary, parental guidance.

### 3.3. Current limitations

The **MIRAGE** method does exhibit some notable weaknesses and biases when applied to NSD-Imagery. The median and worst case reconstructions from **MIRAGE** and other methods can be seen in Appendix A.4 in S1 Text, and in particular we

note how the method is heavily influenced by the prior of the NSD training dataset. Objects that are more well represented in NSD, such as surfers, tend to be reconstructed much more successfully than objects more sparsely observed, such as birds. This is seen even more clearly on the simple stimuli, which are not represented at all in NSD (which contains only natural scenes), and so MIRAGE and other methods tend to produce reconstructions with consistent low-level structural features, but seemingly random semantic content reconstructed in the details of the image.

The NSD-Imagery dataset utilized as a validation set for this work also presents a limitation, as it contains only 18 stimuli, and thus does not allow for large-scale model training or fine-tuning of existing vision decoding models, necessitating a cross-decoding approach. Future models for downstream applications would surely be improved by training or fine-tuning on mental imagery datasets. Such datasets must be a priority for research in this space.

In addition to dataset availability, practical clinical implementation faces the challenge of data acquisition time per subject. Collecting the volume of fMRI data typically required to train robust decoding models is often infeasible for patient populations, however, recent advancements in the "novel subject" problem, such as MindEye2 [2], have demonstrated the potential to fine-tune models on as little as one hour of data by leveraging pre-training on large public datasets. While current data-efficient methods struggle to generalize to mental imagery, we anticipate that future research will bridge this gap, and our results suggest a linear backbone with multimodal guidance to be a promising direction for future subject-adaptive mental imagery decoding.

Currently, the computational requirements to run these models are also an obstacle to realizing medical applications. We trained our ridge regression modules on computing hardware with 512GB of RAM and performed inference for our models using an NVIDIA A100 with 40GB of VRAM. The requirement of such hardware limits the use of our method to researchers with considerable compute resources.

All of these limitations we believe to be rich ground for future work, and we look forward to follow-up research addressing them in more detail.

## 4. Methods

### 4.1. NSD-imagery dataset

To evaluate model performance on internally generated visual representations, we utilize the NSD-Imagery dataset, an extension of the Natural Scenes Dataset (NSD) [25] where the same eight subjects completed a session of additional mental imagery tasks. For a full description of the dataset, please see [1]. This dataset consists of high-resolution 7T fMRI responses collected using identical acquisition protocols to NSD.

**4.1.1. Stimuli.** Prior to scanning, participants in the NSD-Imagery experiment memorized a unique single-letter cue associated with each of the 18 stimuli (Fig 5), which span three distinct categories to assess reconstruction across varying levels of visual and semantic complexity:

(A) *Simple Stimuli:* Six geometric shapes constructed from black bars on a gray background, including four oriented bars (0°, 45°, 90°, 135°) and two crosses ("+" and "×").

(B) *Complex Stimuli:* Five natural scenes selected from the NSD `shared1000` set and one artwork ("The Two Sisters" by Kehinde Wiley), chosen based on recognizability scores from the original NSD sessions.

(C) *Conceptual Stimuli:* Six abstract single-word concepts (e.g., "stripes", "mammal", "banana") rather than specific fixed images. As the vision trials for these concepts involved viewing multiple variant images, we exclude the vision trials for this condition and evaluate only on the imagery trials.

**4.1.2. Task protocol.** The experiment consisted of alternating run types. In *Vision Runs*, participants viewed the target image and its corresponding letter cue for 3 seconds, followed by a 1-second rest. Participants performed a one-back task indicating via button press if the image matched the cue. In *Imagery Runs*, participants were presented only with the

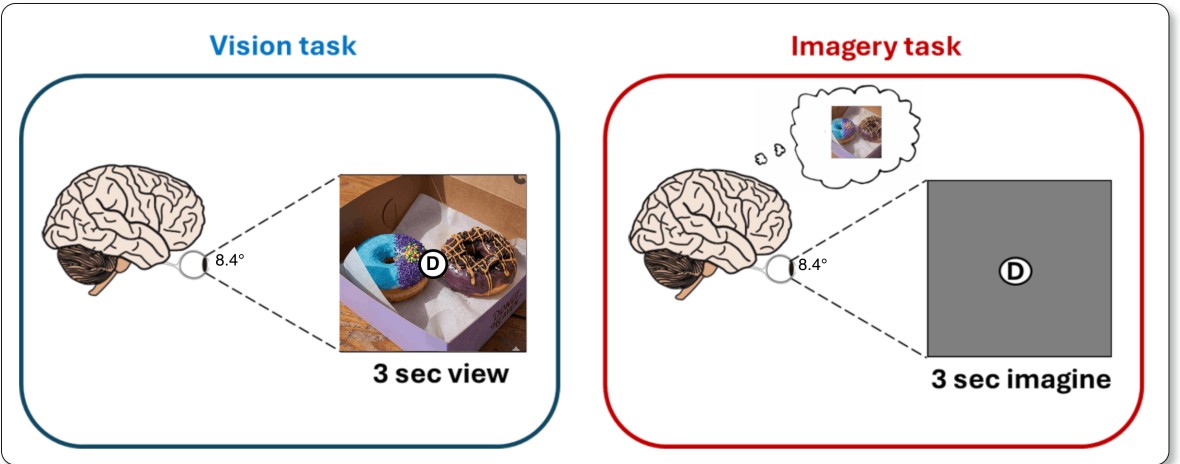

**Fig 5. Overview of the tasks utilized for the NSD-Imagery benchmark.**

letter cue and an empty frame (spanning 8.4° × 8.4° visual angle) for 3 seconds. They were instructed to vividly visualize the corresponding stimulus projected into the frame. This was followed by a 1-second rest and a button-press vividness rating. We note that the NSD-Imagery data were collected in a dedicated scanning session separate from the main NSD experiment. Consequently, **none of the trials in this dataset were used to train MIRAGE** or any other vision decoding models evaluated in this work, which were trained exclusively on the separate visual presentation trials from the NSD core dataset. Furthermore, within the NSD-Imagery session, vision and imagery trials were separated into distinct runs. Vision runs were presented prior to imagery runs to ensure participants correctly recalled the stimuli, but the temporal separation ensures that the decoding of mental imagery is not confounded by hemodynamic responses from preceding visual stimuli. Because there is no session overlaps between the training set (NSD) and the test set (NSD-Imagery), there are also no temporal relationships or block paradigm artifacts for the model to exploit during inference.

### 4.2. MIRAGE

**4.2.1. Datasets.** Our method is trained exclusively on the Natural Scenes Dataset (NSD) [25] which consists of between 22k and 30k fMRI-image pairs per subject (8 subjects). NSD stimuli are sourced from the Common Objects in Context dataset (COCO) [61]. We train models for subjects 1, 2, 5, and 7, as only these subjects completed the full 30k trials of the NSD experiment. We applied the provided nsdgeneral voxel mask at a 1.8 mm resolution to the preprocessed fMRI signals, encompassing numerous visual areas ranging from the early visual cortex to higher-level visual regions. Data from the other 4 NSD subjects are used as a hyperparameter tuning set, as discussed in Section 4.2.3.

To evaluate performance on mental images—our primary decoding target—we apply our method to the NSD-Imagery benchmark discussed in Section 4.1, a small extension of NSD where the same subjects completed mental imagery trials.

**4.2.2. Methodology.** We propose a model, (**MIRAGE**), that respects two important desiderata for decoders that generalize well from vision to mental imagery:

**Reduced model complexity to accommodate the relatively low SNR of mental imagery activity**. Complex, expressive models trained in a high SNR regime may exhibit unacceptably high levels of error variance when tested in a lower SNR setting [62]. Brain activity is known to have much lower SNR during mental imagery than it does during vision, and recent work [1] has found that relatively complex SOTA vision decoding pipelines generalize poorly when applied to mental images. Our decoding pipeline therefore prioritizes robustness over expressivity. In particular, we implemented

a linear ridge regression backbone (Fig 6)—in contrast to the non-linear MLP backbone use more commonly in vision decoding. Ridge regression is effective in high-dimensional, low-SNR settings [66], and is singularly effective for aligning noisy brain activity patterns across individual brains [33].

**Representational alignment to mental images**. Brain activity patterns that represent seen and mental images, respectively, overlap in early visual cortex [7] where structural details of stimuli are encoded, but are most closely aligned in higher-level visual brain areas [13] that are known to represent semantic and/or linguistic aspects of stimuli [67]. Accordingly, our method drives an image generator with the CLS token of a CLIP ViT-L/14 image embedding, and incorporates multi-modal guidance from decoded CLIP ViT-bigG/14 text features (Fig 7).

### 4.2.3. Ridge regression backbone.

To map preprocessed fMRI data to feature representations in our decoding pipeline, we employ parallel $L_2$ regularized ridge regression models trained on each individual feature set. In keeping with our first constraint, ridge regression is chosen because it is known to be effective in low signal-to-noise regimes, in contrast to MLPs that offer advantages for capturing nonlinear relationships, but can also be fragile in brain decoding contexts when the input data are of low SNR. For each set of features, we train a parallel ridge regression model to predict the feature value from our fMRI responses, minimizing the loss function in:

$$\mathcal{L}(\mathbf{w}, \mathbf{b}) = \|\mathbf{Xw} + \mathbf{b} - \mathbf{y}\|_2^2 + \lambda\|\mathbf{w}\|_2^2 \tag{1}$$

where $\mathbf{X} \in \mathbb{R}^{n \times v}$ is the fMRI data matrix of $n$ fMRI trials by $v$ voxels, $\mathbf{y} \in \mathbb{R}^{n \times d}$ is the matrix of target features $d$ for each trial $n$, $\mathbf{w} \in \mathbb{R}^{v \times d}$ is the weight vector mapping fMRI voxels to the feature dimension $d$, $\mathbf{b} \in \mathbb{R}^d$ is the bias vector added to each trial $n$, and $\lambda$ is the ridge parameter controlling the strength of the $L_2$ penalty. To select an optimal value for $\lambda$, we treat the four NSD subjects who are not selected for reconstruction as a hyperparameter tuning set, and perform a grid search over possible values of $\lambda$ on this set to pick the optimal value for generalizing to fMRI responses of mental images. Using this process, we select $\lambda = 100,000$ for all modules of the regression backbone. Details on our hyperparameter search are in Appendix A.1 in S1 Text.

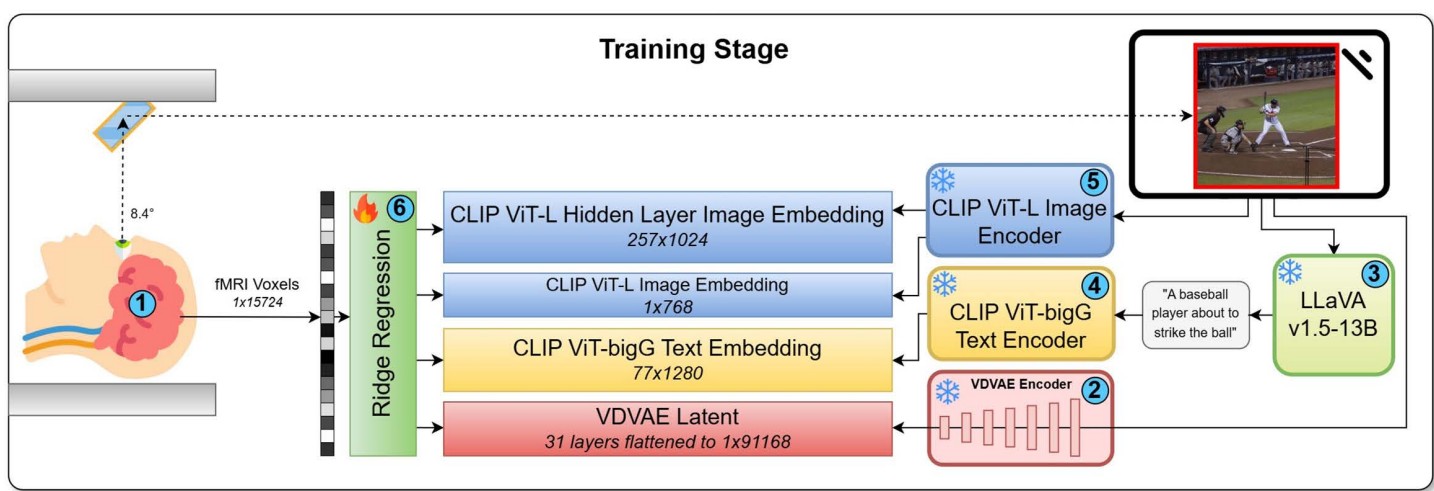

**Fig 6. MIRAGE training pipeline.** (1) Brain activity (7T fMRI) acquired as NSD subjects view > 10K stimuli. (2) Stimuli are passed to VDVAE encoder [37] yielding (1 × 91168) latents (3) LLaVA v1.5-13B [63,64] generates synthetic captions. (4) Captions are encoded into CLIP ViT-bigG/14 text embeddings (77 × 1280) [65]. (5) Stimuli are also passed through the CLIP ViT-L/14 image encoder [23] to generate both CLS token (1 × 768) and hidden layer (257 × 1024) image embeddings. (6) Parallel ridge regression modules are trained from the measured fMRI brain activity to the various feature spaces.

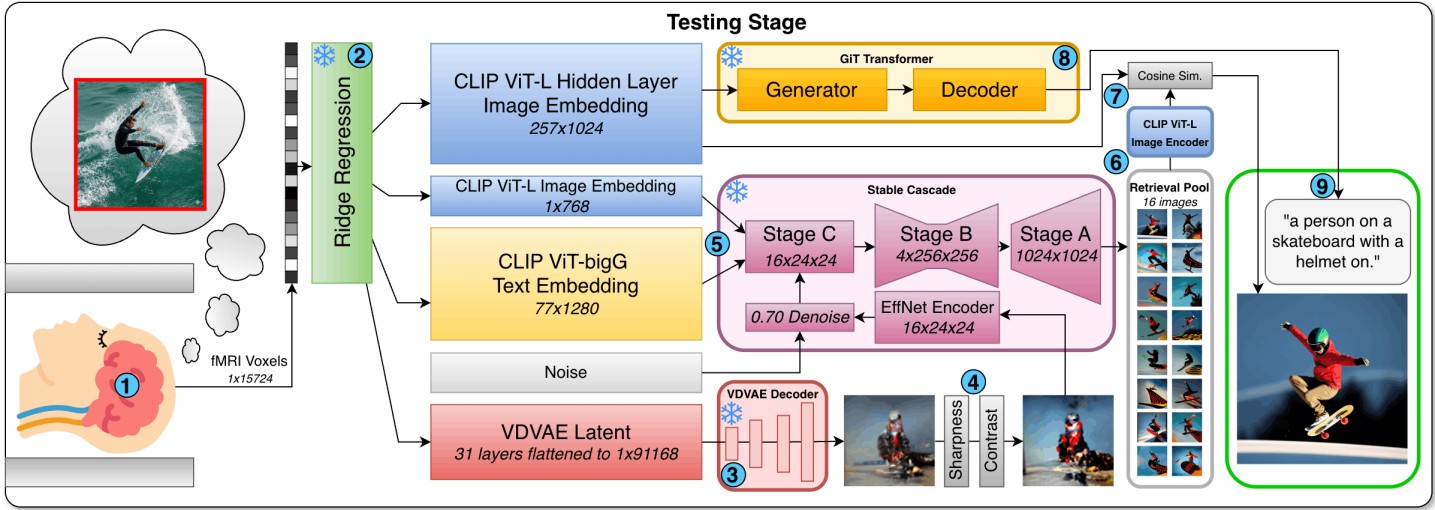

**Fig 7. MIRAGE inference pipeline.** (1) The NSD subjects imagine stimuli from letter cues under 7T fMRI. (2) A set of feature embeddings is predicted by passing the measured fMRI brain activity through our frozen ridge regression models. (3) The VDVAE [37] latents are reconstructed into a low-level image. (4) The image is filtered to boost its structure. (5) The filtered low-level image, decoded image embedding, and decoded text embedding are used as input to Stable Cascade [68] to generate a retrieval pool of 16 candidate reconstructions. (6) Each of the candidate reconstructions is encoded into a CLIP ViT-L/14 hidden layer image embedding [23]. (7) The final reconstruction is automatically selected as the image with the highest cosine similarity to the decoded CLIP ViT-L/14 hidden layer image embedding. (8) The same hidden layer image embedding is passed through the GiT [69] captioning model. (9) A decoded caption is generated along with our final reconstruction.

**4.2.4. Low-level module.** To begin the reconstruction process, we decode and reconstruct a "low-level" image that captures the structural layout of the target image. Inspired by the Brain Diffuser method [28], our approach utilizes a Very Deep Variational Autoencoder (VDVAE) [37] for this task. During training, images are fed into the encoder of the VDVAE to extract latent variables from the first 31 layers of the VDVAE model, which are concatenated into a 1×91168 dimensional feature vector. In the testing phase, we predict these features using our ridge regression backbone and pass them through the VDVAE decoder to generate reconstructed images at 64×64 pixel resolution. Details about our implementation of the VDVAE are in Appendix A.2 in S1 Text.

We observed that when decoding low-level images from fMRI responses to mental images, often the low-level reconstructions have significantly deteriorated features and reduced contrast. To counteract this and boost the influence of our structural layout on subsequent stages of the reconstruction process, we apply a set of image filters to boost the sharpness and contrast of the low-level images. These reconstructions serve as an initial estimate for the diffusion model employed in the subsequent stage of our pipeline.

**4.2.5. Guidance features. Image Features:** Noting that the large high-dimensional hidden layer CLIP ViT-bigG/14 image embeddings used in the MindEye2 architecture (257×1664) fail to robustly drive image generators when decoded from mental images, we use only the CLS token of the CLIP ViT-L/14 model (1×768) for our image embeddings. We hypothesize that the removal of the spatial patch tokens used in the hidden layer of ViT embeddings increases the alignment of our features with the encoding of mental images in the brain.

**Text Features:** Motivated by recent findings suggesting that representations in higher-level visual cortex are well-modeled by language-supervised models [3], we incorporate multi-modal guidance into our method through the addition of an unpooled CLIP ViT-bigG/14 text embedding (77×1280). We hypothesize that this semantic guidance helps bridge the gap between the visual specificity of seen images and the more abstract, concept-driven nature of mental imagery. In

practice, we do not observe the same performance dropoff with high-dimensional text features as we do with high-dimensional image features for driving the diffusion model.

**Recaptioning:** To increase the quality of our decoded text features, we replace the COCO captions for the training set with synthetic captions generated by LLaVA v1.5-13B. Each of the original training captions from the COCO dataset is short (average length of only 10 words). We improve the downstream training performance of our model by generating our own captions that are longer and more descriptive.

### 4.2.6 Reconstruction.

**Diffusion Architecture:** To generate the final reconstructions, we utilize Stable Cascade, one of the latest multi-modal diffusion models by Stability AI. Stable Cascade uses Würstchen [68], a text-to-image architecture that also has support for CLIP Image guidance. This architecture consists of three stages, which separately train an image reconstruction module as a VQ-GAN, a latent image diffusion decoder, and a text/image-conditional latent image diffusion generator. By decoupling the text/image condition from the rest of the generation and decoding, the model learns robust representations and how to translate them into images. This architecture accepts both our small $1 \times 768$ CLIP ViT-L/14 image embeddings, $77 \times 1280$ CLIP ViT-bigG/14 text embeddings, and a partially diffused input image for img2img mode [70]. The model's multi-modal guidance capabilities, native img2img support, and robust multi-stage architecture make it well-suited for our mental imagery reconstruction task.

**Retrieval Pooling:** Because diffusion models are stochastic generators, the quality of the reconstructions they yield will vary from sample to sample. Our pipeline therefore includes a retrieval step to select the best reconstruction from among a small pool samples output by the diffusion model. We decode a CLIP ViT-L/14 hidden layer image embedding $(257 \times 1024)$ that has been unit normalized to approximate a cosine similarity loss in the regression training stage. We then generate 16 candidate reconstructions to form a retrieval pool, pass each candidate through the CLIP ViT-L/14 image encoder to get hidden layer image embeddings, and select only the highest-scoring reconstruction via cosine similarity with our decoded retrieval embedding as our output. We note that using a high-dimensional image embedding for this retrieval operation does not come with the same drawbacks as using a high-dimensional image embedding to drive the image generator, and we observe our process of using a cosine similarity comparison to differentiate between image embeddings to be robust even when the embeddings are decoded from brain activity responses to mental images.

**Caption Decoding:** We implement a caption decoding module that provides a text description of the visual content being decoded. To decode captions, we reuse the CLIP ViT-L/14 hidden layer image features decoded in the retrieval module and pass them through a frozen GiT [69] transformer (which contains a generator and a decoder) to generate a predicted caption, an approach originally pioneered by [71] and MindEye2 [72].

## Supporting information

**S1 Text. Fig A**: Hyperparameter logarithmic grid search over possible values of $\lambda$ for use in Equation 1 (Section 4.2.3). Metrics are the normalized average of all metrics in Table 1 of the manuscript, with imagery performance on the Y axis and vision on the X axis.**Fig B**: Qualitative comparison of reconstruction methods on stimuli seen during the vision trials of NSD-Imagery. Samples selected are the best scoring according to the reconstruction metrics in Table 1 of the manuscript. **Fig C**: Median-case vision reconstructions from the vision trials of NSD-Imagery. Samples selected as median scoring based on metrics in Table 1 of the manuscript. **Fig D**: Median-case imagery reconstructions from the imagery trials of NSD-Imagery. Samples selected as in Fig C. **Fig E**: Worst-case vision reconstructions from the vision trials of NSD-Imagery. Samples selected as lowest scoring based on metrics in Table 1 of the manuscript. **Fig F**: Worst-case imagery reconstructions from the imagery trials of NSD-Imagery. Samples selected as in Fig E. **Fig G**: Best-case vision reconstructions (additional methods) from vision trials of NSD-Imagery. Samples selected as highest scoring based on metrics in Table 1 of the manuscript. **Fig H**: Best-case imagery reconstructions (additional methods) from imagery trials of NSD-Imagery. Samples selected as in Fig G. **Fig I**: Median-case vision reconstructions (additional methods) from vision trials of NSD-Imagery. Samples selected as median scoring based on metrics

in Table 1 of the manuscript. **Fig J**: Median-case imagery reconstructions (additional methods) from imagery trials of NSD-Imagery. Samples selected as in Fig I. **Fig K**: Worst-case vision reconstructions (additional methods) from vision trials of NSD-Imagery. Samples selected as lowest scoring based on metrics in Table 1 of the manuscript. **Fig L**: Worst-case imagery reconstructions (additional methods) from imagery trials of NSD-Imagery. Samples selected as in Fig K. **Table A**: Standard error measurements for evaluation metrics of fMRI-to-Image reconstruction models evaluated on both the vision and mental imagery trials of NSD-Imagery. Values correspond to the standard error spread of values in Table 1 in the manuscript. **Table B**: Quantitative comparison between reconstruction methods on the NSD Shared1000 Test Set. Metrics are the same as Table 1 of the manuscript. **Table C**: Quantitative comparison between reconstruction methods for both imagery and vision trials on simple stimuli. Metrics are the same as Table 1 of the manuscript. **Table D**: Quantitative comparison between reconstruction methods for both imagery and vision trials on complex stimuli. Metrics are the same as Table 1 of the manuscript. **Fig M**: Performance of **MIRAGE** and other methods when averaging across brain activity responses to multiple trial repetitions of the same stimulus. Y-axis is the normalized average of all metrics in Table 1 of the manuscript, X-axis is the number of averaged trial repetitions. **Fig N**: Performance of **MIRAGE** and other methods on NSD-Imagery for Subject 1 when trained on different numbers of fMRI sessions present in NSD. Each session includes approximately one hour of fMRI data. Metrics are the normalized average of all metrics in Table 1 of the manuscript, with imagery performance on the Y axis and vision on the X axis. Methods are indicated by color, with the number of training sessions indicated by the numbers in each dot. **Fig O**: Examples of reconstructions provided at different diffusion strength parameters, images are the ground truth (outlined in red) and reconstructions provided at 0.4, 0.6, 0.8, and 1.0 diffusion strength respectively. **Fig P**: Human identification accuracy of **MIRAGE** (with no CLIP-Image guidance) as a function of diffusion model strength for imagery trials (orange line), vision trials (green line), and a control experiment that used the features directly from the ground truth image and caption (blue line). A dashed line is placed at the 50% chance threshold. Results are from a behavioral experiment that is identical to Experiment 1 (Fig 3), but varied across strength parameters. **Table E**: Quantitative comparison between **MIRAGE** and two Top-1 Retrieval baselines (pooled and hidden layer CLIP ViT-L/14 embeddings), separated by simple and complex stimuli and averaged across all subjects. Metrics are the same as Table 1. Bold indicates the best performance between **MIRAGE** and the retrieval baselines within each stimulus category, and underlines indicate second-best. **Fig Q**: **Top-K retrieval performance vs. pool size for Subjects 1, 2, 5, and 7.** Accuracy (y-axis) is evaluated across varying distractor pool sizes (x-axis) for both mental imagery (left: **A, C**) and vision trials (right: **B, D**). The top row (**A, B**) evaluates retrieval in the pooled ViT-L/14 image embedding space used to drive the MIRAGE generative model, while the bottom row (**C, D**) uses the hidden layer ViT-L/14 space utilized in the retrieval pooling step (Section 4.2.6). Curves denote top-1, top-5, and top-10 performance for simple (light lines) and complex (dark lines) stimuli, with chance levels indicated by corresponding dotted lines. To calculate accuracy, the ground-truth NSD-Imagery stimulus is shuffled with $N$ random distractor images from the NSD shared1000 pool; a success is recorded when the target image ranks within the top $K$ closest matches to the subject's brain-predicted embedding. All curves are bootstrapped across 100 randomly sampled distractor pools for each value of $N$. **Fig R**: An example of the 2 alternative forced choice task used in the first behavioral experiment performed by human raters. **Fig S**: An example of similarity score task used in experiments 2 and 3 of the behavioral experiment performed by human raters. **Appendix A.17**: AI-Generated Images and Copyright Compliance
(PDF)

## Acknowledgments

We would like to thank the many collaborators in the Medical AI Research Center (MedARC) who supported and contributed to the project.

## Author contributions

**Conceptualization:** Reese Kneeland, Thomas Naselaris.

**Data curation:** Reese Kneeland, Thomas Naselaris.

**Formal analysis:** Reese Kneeland, Cesar Kadir Torrico Villanueva, Tong Chen, Jordyn Ojeda, Shubh Khanna, Thomas Naselaris.

**Funding acquisition:** Thomas Naselaris.

**Investigation:** Reese Kneeland, Cesar Kadir Torrico Villanueva, Shubh Khanna, Jonathan Xu, Thomas Naselaris.

**Methodology:** Reese Kneeland, Cesar Kadir Torrico Villanueva, Tong Chen, Thomas Naselaris.

**Project administration:** Paul S. Scotti, Thomas Naselaris.

**Resources:** Thomas Naselaris.

**Software:** Reese Kneeland, Cesar Kadir Torrico Villanueva, Tong Chen, Jordyn Ojeda, Shubh Khanna.

**Supervision:** Reese Kneeland.

**Validation:** Reese Kneeland, Tong Chen, Jonathan Xu, Paul S. Scotti.

**Visualization:** Reese Kneeland, Cesar Kadir Torrico Villanueva, Jordyn Ojeda, Shubh Khanna, Jonathan Xu.

**Writing – original draft:** Reese Kneeland, Cesar Kadir Torrico Villanueva, Tong Chen, Jordyn Ojeda, Paul S. Scotti, Thomas Naselaris.

**Writing – review & editing:** Reese Kneeland, Thomas Naselaris.

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
