## [Decision Letter · Decision Letter 0]

14 Nov 2025

PCOMPBIOL-D-25-01665

MIRAGE: Robust multi-modal architectures translate fMRI-to-image models from vision to mental imagery

PLOS Computational Biology

Dear Dr. Kneeland,

Thank you for submitting your manuscript to PLOS Computational Biology. After careful consideration, we feel that it has merit but does not fully meet PLOS Computational Biology's publication criteria as it currently stands. Therefore, we invite you to submit a revised version of the manuscript that addresses the points raised during the review process.

We look forward to receiving your revised manuscript.

Kind regards,

Yuanning Li

Academic Editor

PLOS Computational Biology

Lyle Graham

Section Editor

PLOS Computational Biology

**Journal Requirements:**

4) Figures 1, 2, 3, 4, 8, 9, 10, 11, 12, 13, 14, 15, 16, 17, 18, 22, and 23. include image of an identifiable person. Please provide written confirmation or release forms, signed by the subject(s) (or their guardian), giving permission to be photographed and to have their images published under a Creative Commons license. You may upload permission forms to your submission file inventory as item type 'Other'. Otherwise, we kindly request that you remove the photograph.

Potential Copyright Issues:

i) Please confirm (a) that you are the photographer of Figures 1, 2, 3, 4, 8, 9, 10, 11, 12, 13, 14, 15, 16, 17, 18, 22, and 23., or (b) provide written permission from the photographer to publish the photo(s) under our CC BY 4.0 license.

**Reviewers' comments:**

Reviewer's Responses to Questions

**Comments to the Authors:**

Reviewer #1: In this paper, a novel model for image reconstruction of mental imagery is proposed: Mental Image Reconstruction using Advanced Generative models (MIRAGE). This model takes into account knowledge about the neural representations of mental imagery to increase reconstruction performance relative to models that are purely trained on perception. For example, mental imagery representations tend to be lower SNR and contain less high spatial frequency info. The authors demonstrate that this model outperforms other models in terms of both standard reconstructions metrics as well as human ratings.

In general, I found that the manuscript presented an innovative new method in a convincing way. The writing style was clear, although quite dense at times. The figures were generally good, although it was not always clear exactly what the take away of a specific figure should be. Given that this method is outside my expertise, I cannot comment too much on the technical details. Therefore, I mostly focus my comments on conceptual clarity.

The authors state that their work is “a necessary step towards applications of mental image reconstruction, including diagnostic instruments for psychiatric conditions [63] and disorders of consciousness [64–66], and alternative communication methods for patients..”. However, it is not immediately obvious to me why mental image reconstruction would be helpful in these cases. Why is it useful to be able to reconstruct an image specifically rather than decode the merely the concepts somebody is thinking about? Some discussion on exactly what the imagistic element would add would be helpful.

Relatedly, how viable is it to train these models in patients, given the large amount of data that is required? Adding some discussion on the practical implementations would be helpful.

I struggled with interpreting the similarity ratings between imagery and vision reconstructions. Why is it interesting to know whether participants rate the reconstructions as similar or not?

For the influence of ablations, only qualitative comparisons in model performance are made. How can we be sure that decreases in performance are specifically due to these ablations and not due to e.g. randomness in the model initiation, reduction of complexity, or anything similar?

Minor comments:

- What does the ‘similarity score advantage’ in experiment 3 mean exactly?

- It would be helpful to start the discussion section with a one or two sentence summary of the main question that was addressed.

Reviewer #2: The paper presents MIRAGE, a multi-modal framework for translating fMRI activity into visual reconstructions, emphasizing robustness across vision-to-imagery transfer. The topic is timely and relevant, addressing the gap between neural decoding of perceived images and imagined images. The paper demonstrates careful engineering and extensive evaluation, but several methodological and conceptual aspects require further clarification or validation. The contribution is empirically valuable, yet conceptually incremental, as it mostly integrates existing modules rather than introducing fundamentally new algorithmic principles.

Pros:

1. The paper tackles an important and challenging question — how to decode mental imagery from fMRI signals by leveraging models trained on visual perception data. This direction is highly relevant for both computational neuroscience and brain–machine interface research, and the authors address it in a systematic way.

2. MIRAGE demonstrates clear improvements over existing methods on the NSD-Imagery dataset, supported by both quantitative metrics and large-scale human evaluations involving around 500 participants. The use of both objective and behavioral measures makes the results more convincing than many prior works.>

3. The paper presents extensive experiments and ablation studies, covering architectural choices, embedding sizes, caption strategies, and multimodal guidance. This systematic approach helps readers understand the contribution of each design component.

4. The decision to use a linear ridge regression backbone instead of deeper MLPs is well justified, given the low signal-to-noise ratio of imagery-related brain data. The integration of low-level VDVAE features and compact CLIP embeddings is also a sound and efficient engineering choice.

Cons:

1. Although the paper presents a well-constructed system, most components--ridge regression, CLIP embeddings, diffusion priors, and caption guidance--are known techniques. The main novelty lies in how they are combined and analyzed rather than in a fundamentally new algorithmic idea. The framing of the contribution should therefore emphasize empirical insight rather than architectural innovation.

2. Because diffusion models can strongly bias the reconstruction toward semantically plausible outputs, it’s unclear how much of MIRAGE’s performance actually reflects brain decoding rather than the model prior. The paper briefly acknowledges this in the appendix, but no systematic control experiments (e.g., with shuffled or random features) are provided. This issue is central to the scientific interpretation of the results.

3. The NSD-Imagery dataset is small (18 stimuli), which makes statistical confidence especially important. However, the paper reports only standard errors and no statistical tests or effect sizes. Similarly, human ratings are reported as averages without inter-rater reliability analyses. More rigorous statistics would substantially strengthen the empirical claims.

4. The ridge coefficient (λ=100,000) and the use of synthetic long captions (from LLaVA) are fixed design choices without justification of their robustness. Showing sensitivity analyses or ablations on these parameters would make the conclusions more solid.

5. Several baselines are reproduced by the authors, while others use numbers reported in previous papers. It’s not fully clear whether all methods were evaluated under consistent conditions. Given that reconstruction quality can vary with sampling strategy or post-processing, this could bias the comparison.

6. The authors suggest that text features help because higher-level brain regions in imagery are “language-like.” This is an interesting hypothesis but not directly supported by voxel-wise or region-based analyses. The paper would benefit from more evidence linking computational results to neural representations.

7. While the ablation studies are good, the discussion of why certain factors help remains superficial. For instance, it would be useful to interpret the improvements from text guidance or reduced embedding dimensionality in light of known properties of visual or semantic cortex.

8. The paper claims that the model "translates fMRI-to-image models from vision to mental imagery," yet it is unclear whether the observed success truly reflects cross-domain generalization or merely robustness to distribution shift. Since both training and testing are within the same dataset family (NSD and NSD-Imagery), a stronger demonstration on unseen subjects or imagery tasks would make this claim more convincing.

9. While comparisons are made against other deep neural decoders, there are no references to simpler baselines such as representational similarity mapping or direct retrieval from a large image-text database using fMRI features. This omission makes it difficult to assess whether MIRAGE’s complexity is actually necessary.

10. The behavioral evaluation is impressive in scale but lacks transparency in design. The manuscript does not specify how stimuli were randomized, whether raters were blind to conditions, or how consistency was ensured across trials. Without these details, it is difficult to judge the robustness of the behavioral conclusions.

11. Several metrics used (e.g., CLIP-based similarity) are inherently aligned with MIRAGE’s feature space, since the model itself uses CLIP embeddings for decoding. This could give MIRAGE an advantage compared with baselines that use different representational spaces, potentially inflating performance estimates.

12. The discussion links MIRAGE’s improvements to neurocognitive mechanisms of imagery (e.g., semantic and linguistic engagement). However, no direct neural evidence supports this explanation—no ROI-based correlation, temporal analysis, or representational similarity analysis is presented. These interpretations, while interesting, remain speculative.

13. The paper does not systematically discuss failure modes — for example, whether MIRAGE tends to hallucinate specific object types, blur boundaries, or misrepresent spatial layouts. Without such an analysis, readers cannot fully understand the model’s limitations or potential biases.

Other issues and suggestions:

1. The abstract and introduction occasionally overstate originality. Phrases like “first robust multimodal architecture” could be rephrased as “a systematically evaluated multimodal framework for imagery decoding.”

2. Run a small-scale control where brain-decoded features are replaced with shuffled or averaged features, to quantify the prior’s contribution.

3. Show examples at different guidance strengths to illustrate when reconstructions are driven by brain signals versus prior bias.

4. Show how performance changes with λ and embedding size to confirm robustness.

5. Indicate the number of samples averaged in each figure and ensure all key experimental details (e.g., λ grid search) are referenced in the main text.

6. Some other grammar errors.

Reviewer #3: This paper presents an algorithm that can use an fMRI-to-image model

trained on people looking at actual images and use it to visualize

something related to visual mental imagery. The work is very

interesting and the ablation studies nicely demonstrate the importance

of different aspects of the model. Including human evaluations is also

a strength.

My main criticism with the paper is that it does not describe the

dataset (and the protocol for recording it) in enough detail to

ascertain whether the results are meaningful as brain decoding as

opposed to representing something else (for example representing

"time" as has been shown to be the case in studies with block design

where training and testing items of the same class occur close in

time). The paper should be self-contained and contain these details

critical for evaluating the relevance of the work. (I should mention

that I did look at the NSD-Imagery referenced CVPR paper by

overlapping authors which also does not describe the experiment as

clearly as needed for a contribution to Neuroscience).

A description of the experimental design and details of collection of

the NSD and NSD-Imagery is critical. Especially with recurring issues

in this field (where people have used datasets with block-design), it

is critical that the dataset and experimental design be fully

characterized and included in the paper.

Similarly, details are missing in the processing steps of the

algorithm. Authors say "we apply a set of image filters to boost the

sharpness and contrast of the low-level images", but no detail of

these filters are given. Please give all the details for this

processing. What image filters do you use to "boost the sharpness and

contrast of the low-level images" How are any hyperparameters in this

process chosen?

The authors acknowledge that the standard metrics may not be reliable

indicators of quality. Including human evaluations is great, but what

about also looking at top-1 and top-5 retrieval accuracies?

How do you compute the "normalized feature metric performance" when

some of the metrics are better larger and some better smaller. Please

give the equation. Are median and worst case reconstructions based

on this normalized metric?

While I agree that any experiments on patients and participants in

general should be done with proper consent and IRB oversight, I do not

think that fMRI recording should be defined as an invasive medical

procedure. Any such statements should not be made lightly but by

proper committees trained in these matters.

**Have the authors made all data and (if applicable) computational code underlying the findings in their manuscript fully available?**

Reviewer #1: None

Reviewer #2: **No:** I logged into the code link provided by the authors, but it shows an error message. Therefore, I am unable to determine whether the authors have provided the complete data and code.

Reviewer #3: **No:** The code is available. The data says it is available and might be but the form accessed through their site

"If you would like to access the NSD dataset, please fill out this short NSD Data Access Agreement." https://docs.google.com/forms/d/e/1FAIpQLSduTPeZo54uEMKD-ihXmRhx0hBDdLHNsVyeo_kCb8qbyAkXuQ/viewform

doesn't currently mention the NSD Imagery component

So I am not sure if the data is available or not.

PLOS authors have the option to publish the peer review history of their article (what does this mean?). If published, this will include your full peer review and any attached files.

Reviewer #1: No

Reviewer #2: **Yes:** Lingxiao Yang

Reviewer #3: No

**Figure resubmission:**
---

## [Decision Letter · Decision Letter 1]

21 Apr 2026

PCOMPBIOL-D-25-01665R1

MIRAGE: Robust multi-modal architectures translate fMRI-to-image models from vision to mental imagery

PLOS Computational Biology

Dear Dr. Kneeland,

Thank you for submitting your manuscript to PLOS Computational Biology. After careful consideration, we feel that it has merit but does not fully meet PLOS Computational Biology's publication criteria as it currently stands. A few outstanding questions need to be properly addressed before publication. Therefore, we invite you to submit a revised version of the manuscript that addresses the points raised during the review process.

We look forward to receiving your revised manuscript.

Kind regards,

Yuanning Li

Academic Editor

PLOS Computational Biology

Lyle Graham

Section Editor

PLOS Computational Biology

**Journal Requirements:**

1) Your manuscript's sections are not in the correct order.  Please amend to the following order: Abstract, Introduction, Results, Discussion, and Methods

**Reviewers' comments:**

Reviewer's Responses to Questions

**Comments to the Authors:**

Reviewer #1: The authors have addressed my previous comments.

Reviewer #2: The authors have addressed most of the concerns raised in the previous round. The revised manuscript includes several additions: a retrieval baseline comparison (Appendix A.13), controlled experiments on diffusion prior strength (Appendix A.12), clarified dataset design details ([Sec sec007]), and adjustments to novelty claims. The core contribution - identifying architectural choices that improve cross-domain generalization from seen to mental imagery - is now presented more as empirical insight. However, a few issues remain and should be corrected before acceptance.

1. In Section 6.2 (Societal Impact, lines 441–444), the manuscript states: “We propose that when deployed in a clinical setting, brain decoding should be defined as an invasive medical procedure…” In the response to Reviewer #3, the authors agreed that fMRI acquisition is physically non-invasive. These two statements are inconsistent. Please revise the main text, for example by removing the word “invasive” or rephrasing as “a medical procedure that yields private health information (even though fMRI acquisition itself is non-invasive).”

2. In Section 5.3 and Appendix A.12, the authors state that “reconstructions with zero guidance (pure VDVAE decoding) remained highly identifiable.” However, Figure 23 shows diffusion strength parameters only from 0.4 to 1.0, with no strength = 0 condition. Please clarify where the zero-guidance results are reported, or add a strength = 0 data point to the figure.

3. The response to Reviewer #1 regarding data-efficient fine-tuning (MindEye2) was thoughtful. Consider adding a forward-looking statement in Section 6.3 (Limitations), for example: “While current data-efficient models struggle with mental imagery, our results suggest that a linear backbone with multimodal guidance may be a promising direction for future subject-adaptive mental imagery decoding.”

4. In Appendix A.13, the authors claim that MIRAGE “substantially outperforms both Top-1 retrieval baselines across the majority of quantitative metrics.” While this holds for semantic and high-level metrics, Table 6 shows that retrieval baselines are competitive or better on some low-level metrics (e.g., SSIM for complex stimuli). Please revise the conclusion to be more balanced, for example: “MIRAGE outperforms retrieval baselines on most high-level and semantic metrics, while retrieval remains competitive on certain low-level metrics.”

**Have the authors made all data and (if applicable) computational code underlying the findings in their manuscript fully available?**

Reviewer #1: Yes

Reviewer #2: Yes

PLOS authors have the option to publish the peer review history of their article (what does this mean?). If published, this will include your full peer review and any attached files.

**Do you want your identity to be public for this peer review?** For information about this choice, including consent withdrawal, please see our Privacy Policy.

Reviewer #1: No

Reviewer #2: No

**Figure resubmission:**
---

## [Editor Report · Decision Letter 2]

23 Apr 2026

Dear Mr. Kneeland,

We are pleased to inform you that your manuscript 'MIRAGE: Robust multi-modal architectures translate fMRI-to-image models from vision to mental imagery' has been provisionally accepted for publication in PLOS Computational Biology.

Best regards,

Yuanning Li

Academic Editor

PLOS Computational Biology

Lyle Graham

Section Editor

PLOS Computational Biology

---

## [Editor Report · Acceptance letter]

PCOMPBIOL-D-25-01665R2

MIRAGE: Robust multi-modal architectures translate fMRI-to-image models from vision to mental imagery

Dear Dr Kneeland,

I am pleased to inform you that your manuscript has been formally accepted for publication in PLOS Computational Biology. Your manuscript is now with our production department and you will be notified of the publication date in due course.

With kind regards,

Anita Estes
